# Crosstalk between Noncoding RNAs and the Epigenetics Machinery in Pediatric Tumors and Their Microenvironment

**DOI:** 10.3390/cancers15102833

**Published:** 2023-05-19

**Authors:** Anup S. Pathania

**Affiliations:** Department of Biochemistry and Molecular Biology & The Fred and Pamela Buffett Cancer Center, University of Nebraska Medical Center, Omaha, NE 68198, USA; anup.pathania@unmc.edu

**Keywords:** non-coding RNAs, epigenetics, pediatric tumors

## Abstract

**Simple Summary:**

Non-coding RNAs (ncRNAs) are functional RNA molecules that occupy a large fraction of the genome but are not translated into proteins. However, they serve key regulatory roles in the regulation of gene expression at the transcriptional and translational levels. Recent studies have suggested that ncRNAs can control gene activity by regulating epigenetic mechanisms. Epigenetic changes can modify genomic DNA and histones, thus influencing gene activity without altering the DNA sequence. Atypical epigenetic modifications in specific tumor-promoting (oncogenes) and tumor suppressors genes can result in cancer development and growth. Pediatric tumors are known to be more prone to epigenetic changes compared to adult tumors. These changes can significantly impact the development and progression of these tumors. The abnormal expression of ncRNAs has been linked to epigenetic alterations and tumor development in various types of pediatric cancers. This review discusses the role of ncRNAs as critical regulators of epigenetic modifications that can drive the malignant phenotype in pediatric tumors. It also provides insights into the therapeutic significance of ncRNAs in targeting epigenetic alterations for the treatment of pediatric cancers.

**Abstract:**

According to the World Health Organization, every year, an estimated 400,000+ new cancer cases affect children under the age of 20 worldwide. Unlike adult cancers, pediatric cancers develop very early in life due to alterations in signaling pathways that regulate embryonic development, and environmental factors do not contribute much to cancer development. The highly organized complex microenvironment controlled by synchronized gene expression patterns plays an essential role in the embryonic stages of development. Dysregulated development can lead to tumor initiation and growth. The low mutational burden in pediatric tumors suggests the predominant role of epigenetic changes in driving the cancer phenotype. However, one more upstream layer of regulation driven by ncRNAs regulates gene expression and signaling pathways involved in the development. Deregulation of ncRNAs can alter the epigenetic machinery of a cell, affecting the transcription and translation profiles of gene regulatory networks required for cellular proliferation and differentiation during embryonic development. Therefore, it is essential to understand the role of ncRNAs in pediatric tumor development to accelerate translational research to discover new treatments for childhood cancers. This review focuses on the role of ncRNA in regulating the epigenetics of pediatric tumors and their tumor microenvironment, the impact of their deregulation on driving pediatric tumor progress, and their potential as effective therapeutic targets.

## 1. Introduction

The genomic landscapes of pediatric cancers are different from adult cancers. A higher prevalence of multiple genetic alterations and structural variants that drive cancer progression is present in adult tumor genomes. In contrast, pediatric tumor genomes share a low mutational burden and fewer structural variations. Gröbner and colleagues identified single-gene driver mutations in 57% and germline mutations in 7.6% of pediatric cancers [1]. Additionally, the authors found less overlap between driver mutations of pediatric and adult tumors, suggesting that different genes are mutated in these cancers. Despite low mutation frequency, pediatric cancers often grow faster than adult ones. This makes them diagnosed at a more advanced stage when the tumor has already spread to other body parts. Epigenetic alterations that modify gene activity without changing the DNA sequence play a central role in the initiation, progression, along with the acquisition of a highly invasive phenotype in most pediatric tumors [2,3,4]. These alterations are due to the presence of cancer-driving point mutations in genes that code for proteins involved in epigenetic regulation. Another common feature of some pediatric cancers is the presence of fusion oncogenes, which can drive tumor development [5]. A fusion gene is a hybrid gene composed of two previously independent genes that can occur due to chromosomal rearrangements such as chromosome translocation, insertion, deletion, tandem duplications, or inversion. The resulting fusion genes are either oncogenic or truncated tumor suppressors that have lost their functions, resulting in enhanced cell division and cell proliferation [6].

Furthermore, the dynamic and complex microenvironment around tumor cells, composed of the extracellular matrix, the tumor stroma, blood vessels, and infiltrating immune and inflammatory cells, plays a significant role in supporting tumor growth and progression. The microenvironment in pediatric solid tumors is enriched in myeloid cells and macrophages and has fewer T cells due to lower antigen presentation on tumor cells [7,8,9]. Therefore, strategies to target myeloid cell populations for therapeutic benefits are an active area of research in pediatric tumors. Overall, the complex nature of pediatric tumors, comprising epigenetic alterations, single-gene and germline driver mutations, oncogenic fusion genes, and the dynamic tumor microenvironment, drives cancer growth and poses challenges for targeting tumors therapeutically. Furthermore, the discovery of thousands of ncRNAs over the past decade has added another layer of complexity to the regulation of pediatric tumors. These ncRNAs are upstream regulators of genes involved in epigenetic alterations and cell-type specificity during embryonic differentiation [10,11]. They also regulate oncogenic gene fusions [12] and exchange information between tumors and various cell types in the tumor microenvironment [13].

The main classes of ncRNAs include microRNAs (miRNAs), long noncoding RNAs (lncRNAs), small interfering RNAs (siRNAs), small nuclear RNAs (snRNAs), circular RNAs (circRNAs), and PIWI-interacting RNAs (piRNAs). Among these, miRNAs and lncRNAs have been extensively studied in adult and pediatric tumors in recent years. The deregulation of both types of transcripts has been strongly linked to tumor development and growth in children and adolescents [14]. MiRNAs function by binding to complementary sequences in the 3′ untranslated region (UTR) of target messenger RNA (mRNA) molecules, resulting in either translational inhibition or mRNA degradation. RNAs larger than 200 nucleotides are considered lncRNAs, which can regulate gene expression at different stages, including transcriptional, post-transcriptional, and epigenetic levels. The exact mechanisms via which lncRNAs function depend on the specific lncRNA and the cellular context in which it acts. For instance, lncRNAs can associate with chromatin-modifying complexes to alter the structure and accessibility of chromatin [15]. They can also regulate mRNA stability [16], protein–protein interactions [17], and post-translational protein modifications, such as phosphorylation [18] and ubiquitination [19], thereby influencing gene expression and cellular processes.

Both miRNAs and lncRNAs have emerged as crucial players in cancer epigenetics, with many studies implicating their role in the epigenetic regulation of multiple genes. These ncRNAs regulate various epigenetic processes, including DNA methylation and histone modification, by controlling chromatin-modifying enzymes such as histone or DNA methyltransferases, histone and DNA demethylases, and histone deacetylases. The dysregulation of miRNAs and lncRNAs has profound effects on the expression of these enzymes, which can lead to alterations in the global chromatin and transcriptional landscape in tumors (Figure 1).

Recent studies have shown that miRNAs, lncRNAs, and other ncRNA types play a critical role in epigenetic regulation during cellular differentiation and tissue development [20,21,22,23,24]. Deregulation of these ncRNAs has been strongly linked to epigenetic deregulation of pediatric tumors and plays a critical role in their initiation, progression, and response to therapy [25,26,27]. The growing role of these ncRNAs in the epigenetics of pediatric cancers makes them one of the most promising therapeutic targets for cancer treatment. In addition, the differential expression of ncRNAs between tumors and corresponding normal cells suggests their potential use as diagnostic and prognostic biomarkers in pediatric cancers [28]. This review focuses on miRNAs and lncRNAs, in the context of pediatric tumors, where their deregulation can promote epigenetic alterations. Conversely, the loss of epigenetic control of these ncRNAs can alter their regulation and disrupt the crosstalk between epigenetic machinery and ncRNAs in the tumor microenvironment (TME). This can confer a growth advantage to tumor cells and influence tumor growth, development, invasion, and metastasis. In addition, the roles of two other ncRNAs, PIWI-interacting RNAs (piRNAs) and small nuclear RNAs (snRNAs), are briefly discussed in this context.

## 2. Role of ncRNAS in the Epigenetic Regulation of Pediatric Tumors

### 2.1. Histone Modifications

Histone proteins package DNA into nucleosomes, and modifications in their structures serve as a mechanism to control gene expression. Alterations in the patterns of histone modifications can change nucleosome dynamics, affecting DNA transcription and leading to cancer initiation and development [29]. There is a class of miRNAs known as Epi-miRNAs that regulate histone functions by directly targeting enzymes involved in modifying histones. The best-studied example is the methylation of histone H3 on lysine 27 (H3K27) by enhancer of zeste 2 (EZH2). This modification induces widespread gene repression and is linked to the pathogenesis of many pediatric cancers [30,31,32,33,34,35]. EZH2 is an enzymatic component of polycomb repressive complex 2 (PRC2), a histone-modifying methyltransferase essential for normal embryonic development and stem-cell differentiation [36,37]. PRC2 catalyzes mono-, di-, and trimethylation of histone H3 at lysine 27 (H3K27me1/2/3), causing chromatin compaction and transcriptional repression [38].

#### 2.1.1. miRNAs

Several miRNAs directly target EZH2 and play an important role in regulating EZH2-mediated gene silencing. These miRNAs can alter EZH2 activity and are associated with poor clinical outcomes and therapeutic responses in cancer patients. For instance, miR-101-3p directly binds to the EZH2 3′UTR and inhibits its translation, which is associated with reduced growth in medulloblastoma (MB) tumors [39]. MiR-137 targets the EZH2 3′UTR in neuroblastoma cells, leading to decreased H3K27 methylation in the genomic regions of tumor suppressor genes clusterin (CLU) and nerve growth factor receptor (NGFR). Loss of H3K27 methylation upregulates CLU and NGFR, induces apoptosis, and potentiates resveratrol-induced cell death in neuroblastoma cells [40]. Neuroblastoma arises from disrupted differentiation in neural crest cells that can lead to oncogenic transformation [41]. EZH2-mediated H3K27 trimethylation of neuron-specific genes decreases during neuronal differentiation, resulting in an increase in their expression. MiR-124 plays an essential role in regulating EZH2 levels during neuronal differentiation. MiR-124 downregulates EZH2, which upregulates its target genes that are involved in neural differentiation. Moreover, the efficiency of neuronal differentiation is significantly reduced in embryonal P19 cells that express EZH2 transgene without the miR-124 3′UTR target site, suggesting that miR-124 is required for efficient neuronal differentiation [42].

Another example includes miRNA-mediated regulation of Wilms tumor 1 (WT1) gene functions in the differentiation of mesenchymal stem cells during nephrogenesis. WT1 is a tumor suppressor located at chromosome 11p13 and involved in kidney and gonadal differentiation [43]. Mutations in the WT1 gene are linked to 20% of Wilms tumors, a rare kidney cancer in children that develops during embryogenesis. WT1 interacts with EZH2-5′UTR and RNA-induced silencing complex (RISC) containing miR-26a-5p and miR-101-3p to facilitate miRNA-mediated EZH2 posttranscriptional silencing. This silencing inhibits EZH2-mediated epigenetic repression of differentiation genes in mesenchymal stem cells during nephrogenesis. This suggests the critical role of miR-26a-5p and miR-101-3p in the regulation of cellular differentiation, particularly during embryogenesis [43]. Furthermore, mutations in miRNA processing enzymes DICER1 or DROSHA are linked with Wilms tumor, indicating the crucial role of miRNAs in the differentiation of mesenchymal stem cells during nephrogenesis and the development of Wilms tumors [44,45]. Moreover, an miRNA profiling study of childhood infratentorial ependymomas revealed differentially expressed miRNAs between grade II and III tumors. The study found that miR-17-5p, miR-19a-3p, and miR-106b-5p expressions can effectively differentiate between grade II and III ependymoma tumors and significantly correlate with EZH2 expression [46]. The expression of these miRNAs was significantly lower in grade II than in grade III infratentorial tumors. These findings imply that miR-17-5p, miR-19a-3p, and miR-106b-5p could serve as important biomarkers for differentiating grade II and III infratentorial ependymoma tumors, along with EZH2.

The other components of PRC2 complex embryonic ectoderm development (EED) and SUZ12 are also targeted by miRNAs, which inhibit repressive histone methylation and activate gene transcription [47,48,49,50]. For instance, EZH2 and EED are direct targets of the tumor suppressor miR-101 in hepatocellular carcinoma (HCC) cells [47,48]. Ectopic expression of miR-101 in HCC cells decreases the global expression levels of H3K27me3-marked genes and induces cell-cycle arrest [48]. MiR-101 displays multiple other roles in controlling several cellular processes in these tumors, including survival by suppressing a cohort of oncogenes such as STMN1, JUNB, and CXCR7 [47]. Furthermore, ectopic expression of miR-101 decreases H3K27me3 occupancy on an oncogenic transcription cofactor protein LIM domain-only protein 3 (LMO3) in gliomas [51]. Other miRNAs such as mir-767-5p, miR-105, and miR-200 target SUZ12 to regulate glioma cell proliferation and the formation and maintenance of cancer stem cells [49,50,52]. Specifically, miR-200 levels decrease during cancer stem-cell formation, which leads to an increase in SUZ12 expression and H3-K27me3 of the E-cadherin gene, repressing their activity [50]. Downregulation of E-cadherin expression is associated with enhanced expression of pluripotent genes c-MYC, NESTIN, OCT3/4, and SOX2, and promotes stemness in cancer cells [53].

Overall, these studies suggest that miRNAs are critical in regulating the H3K27me3 repressive mark found on many silenced gene promoters. MiRNA-mediated chromatin alterations can change the genomic landscape of histone modifications and are crucial for maintaining cellular homeostasis. Conversely, the presence of histone methylation marks in the miRNA-harboring genes indicates the existence of regulatory feedback loops that epigenetically regulate miRNA transcription. For instance, EZH2 overexpression induces H3K27me3-mediated silencing of miR-326 and its host gene β-arrestin1, limiting their growth suppressive activity in MB cells [54]. Mir-326 and β-arrestin1 function as tumor suppressors in MB cells by inhibiting EZH2 activity either via mir-326-mediated EZH2 mRNA degradation or via β-arrestin1-induced EZH2 acetylation. EZH2 repression upregulates transcription of its target proapoptotic genes and increases apoptosis in MB cells [54]. This suggests that disturbances in feedback loop regulation between EZH2 and its regulatory miRNAs can confer a selective growth advantage to cancer cells and turn on genes associated with increased tumorigenicity in MB. Another example includes the epigenetic silencing of miR-139 (miR-139-3p and miR-139-5p) by PRC2-mediated hypermethylation, which frequently occurs in mixed-lineage leukemia (MLL)–AF9 fusion protein-expressing acute myeloid leukemia (AML) [25]. MLL1 translocations are found in approximately 10% of all leukemias and are more common in primarily acute lymphoblastic leukemia (ALL) patients younger than 1 year of age at diagnosis [55]. MLL-AF9 recruits PRC2 to the regulatory regions of miR-139 host gene *PDE2A.* PRC2 further recruits RNA polymerase II subunit M (POLR2M), a subunit of a specific form of RNA polymerase II termed Pol II(G), which binds to E1, E2, and the transcriptional start site of MIR139, inhibiting its transcription. Mir139 inhibition enhances the colony-forming capacity and proliferation of MLL-AF9 cells, suggesting its tumor-suppressive functions [25].

Furthermore, activation of the cyclic adenosine monophosphate/protein kinase A (cAMP/PKA) pathway in human glioblastoma cells induces PRC2 mediated H3K27me3 of miR-1275, suppressing its activity. Downregulation of miR-1275 upregulates its target gene, glial fibrillary acidic protein (GFAP), a differentiation marker of mature astrocytes [56]. This suggests that epigenetic modulation of miRNAs is essential to regulate cell differentiation. Deregulation in this process can help cancer cells quickly evolve or diversify their genetics to give them a survival advantage.

#### 2.1.2. LncRNAs

LncRNAs can interact with various types of miRNAs and modulate their expression and mRNA regulation. They can regulate miRNA-mediated gene expression by binding to the miRNA, preventing its interaction with its target, or by epigenetically suppressing its expression. For instance, lncRNA taurine upregulated gene 1 (TUG1) epigenetically suppresses miR-34a expression by recruiting EZH2 to its promoter region. This leads to the inhibition of miR-34a tumor-suppressive functions associated with the development of drug resistance to Adriamycin in AML cells [57]. Another lncRNA, LINC01116, directly interacts with EZH2 and promotes hypermethylation of the miR-424-5p promoter, suppressing its expression. The miR-424-5p downregulation increases its downstream target HMGA2, promoting cancer growth and doxorubicin resistance in osteosarcoma cells [58]. LncRNAs can directly interact with the promoter region of miRNA and affect chromatin accessibility via histone modifications in AML cells [59]. For example, lncRNA HOTTIP binds to miR-196b and increases H3K4me3 and H3K79me2 enrichment in its genomic locus, which comprises HOX cluster miRNAs, including miR-10a, miR-10b, miR-196a-1, miR-196a-2, and miR-196b. Consequently, these miRNAs levels go up, destabilizing their target tumor suppressor mRNAs that lead to enhanced AML tumorigenesis and promote leukemogenesis [59].

Other mechanisms via which lncRNAs function to regulate transcription include enhancing EZH2 mRNA stability or recruiting EZH2 to regulatory regions of target genes, promoting their EZH2-mediated epigenetic silencing. For example, LINC00313 mediates the EZH2 interaction with DNA/RNA-binding protein FUS, which increases EZH2 mRNA stability. The upregulation of EZH2 epigenetically silences its target tumor suppressor gene PTEN and promotes osteosarcoma cell proliferation, growth, migration, and invasion [60]. LncRNA LSINCT5 recruits EZH2 to induce H3K27me3 on the promoter region of the APC gene, an inhibitor of Wnt/-catenin signaling. APC silencing activates the Wnt/β-catenin pathway in osteosarcoma cells, promoting their proliferation, migration, and invasion [61]. Moreover, lncRNA HOXD-AS1 recruits EZH2 to the p57 promoter, epigenetically silences p57 expression, and promotes osteosarcoma growth [62].

Transcriptional repression of genes by lncRNA-mediated epigenetic gene silencing is a crucial mechanism that determines cell fate during development. Deregulation of lncRNA functions can disrupt cell development, differentiation, and homeostasis, leading to malignant progression. For instance, the lncRNA neuroblastoma-associated transcript-1 (NBAT-1) plays an essential role in neuronal differentiation and neurogenesis by activating the neuronal-specific transcription factor NRSF/REST. Concurrently, NBAT-1 interacts with EZH2 and promotes H3K27me3 modification at the promoters of genes involved in cell proliferation and invasion. The loss of NBAT-1 inhibits neural differentiation, promotes neuroblastoma tumor development, and correlates with poor survival rates in neuroblastoma patients [26]. Another lncRNA, SNHG1, is essential for neuroblastoma cell fate determination and controls various cellular processes such as growth, proliferation, differentiation, and apoptosis. SNHG1 interacts with HDAC1/2 and regulates chromatin accessibility for a network of transcription factors (TFs), including PHOX2B, HAND2, GATA3, ISL1, TBX1, and MYCN. These TF networks form the core transcriptional regulatory circuit (CRC) required to initiate and maintain the transformed phenotype in MYCN-amplified neuroblastoma [63,64].

Many transcription factors (TFs) involved in pluripotency maintenance regulate the transcription of lncRNAs to induce genes necessary to preserve the undifferentiated state. One example is octamer-binding transcription factor 4 (OCT-4), which maintains the pluripotency of embryonic stem cells and activates Suv39h1as, an antisense lncRNA to H3K9 di- and trimethylase SUV39H1. LncRNA Suv39h1as mutant embryonic cells showed increased SUV39H1 protein levels and upregulated global H3K9me2 and H3K9me3, resulting in global transcriptional silencing and restricting cell fate by promoting commitment into differentiation [65]. OCT-4 also plays a crucial role in tumor-initiating cell functions in many pediatric tumors [66,67,68,69]. The emerging role of downstream lncRNAs in mediating OCT-4 functions makes them promising therapeutic targets to stop the proliferation of tumor-initiating cells.

Histone ubiquitination and degradation represent another mechanism via which lncRNAs can influence chromatin’s overall structure and regulate gene expression. The alteration of histone ubiquitination is frequently seen in pediatric cancers [70,71,72,73], and recent studies have suggested the pivotal role of lncRNAs in regulating this biological process. One example is lncRNA EPAT (lncEPAT), which binds to deubiquitinase (DUB) USP16 and prevents its chromatin recruitment, blocking USP16-mediated H2A deubiquitination. This, in turn, induces alterations in chromatin structure associated with silencing cell senescence genes, including CDKN1A and clusterin. Inhibition of lncEPAT promotes USP16-induced cell-cycle arrest and cellular senescence in glioblastoma cells [74]. In addition, lncRNAs can directly bind and expose histones to post-translational modifications, aiding the accessibility of the genome to regulatory transcription factors.

For instance, long noncoding enhancer RNA (lnc-eRNA) SEELA binds to the K31 amino acid of histone H4 and enhances the chromatin occupancy of histone modifiers such as BRD4 [75]. SEELA acts as a modular scaffold by interacting with BRD4 and K31 histone H4, which increases the enhancer activity of its target SERINC2. SERINC2 functions as an oncogene in AML and is involved in AML growth and progression [75]. Moreover, HOXA family genes HOXA9/10 increase lncSEELA transcription by promoting transcriptional activation of BRD4 [75]. Dysregulation of HOX genes is associated with pediatric malignancies such as AML and ALL [76,77]. The downstream role of lncRNAs in mediating HOXA gene family functions suggests the importance of these oncogenic lncRNAs as anticancer drug targets in Hoxa-dependent tumors like AML.

Furthermore, lncRNAs such as NEAT1 can regulate gene expression by modulating histone phosphorylation, which occurs on serine, threonine, and tyrosine residues and is associated with transcriptional activation. NEAT1 plays a protective role in preventing excessive DNA damage and regulates p53 activation in response to oncogenic stress. During DNA damage, NEAT1-silenced cells show high expression of γ-H2AX serine 139 phosphorylation and are more sensitive to DNA damage-induced cell death [78]. Similarly, depletion of other lncRNAs such as NIHCOLE and CRNDE in HCC cells showed high γ-H2AX phosphorylation following irradiation, which is associated with low non-homologous end-joining (NHEJ) activity and high DNA damage accumulation [79,80]. These findings suggest that lncRNAs NEAT1, NIHCOLE, and CRNDE can confer malignant adaptation and therapy resistance to cells by modulating H2AX phosphorylation and DNA damage response.

### 2.2. DNA Methylation

DNA methylation involves transferring a methyl group onto the cytosine’s C5 position, forming 5-methylcytosine on DNA. DNA methylation silences gene expression by either preventing the binding of TFs to the gene or recruiting factors involved in gene repression [81]. NcRNAs regulate genes that facilitate DNA methylation, and any misregulation in this process can contribute to tumor development. Conversely, many genomic regions of ncRNAs in tumors show different DNA methylation patterns than their normal counterparts. These DNA methylation changes are frequent in pediatric cancers and closely associated with driver epigenetic events that can promote cancer cell survival and growth. This section focuses on the role of miRNA and lncRNA in the DNA methylation process and the epigenetic mechanisms regulating these ncRNAs to control gene expression in pediatric cancers.

#### 2.2.1. miRNAs

DNA methylation is performed by a family of enzymes known as DNA methyltransferases (DNMTs) that transfer a methyl group from the universal methyl donor, S-adenosyl-L-methionine (SAM), to the DNA [82]. DNMTs are direct targets of many upstream miRNAs that negatively regulate their functions. DNMTs such as DNMT1, DNMT3A, and DNMT3B are direct targets of miR-29b in AML patients. MiR-29b downregulation in AML is associated with hypermethylation and loss of expression of tumor suppressor genes ESR1 and p15 [83,84]. On the other hand, miR-152 targeting DNMT1 induces DNA demethylation in distinct genomic regions of nitric oxide synthase 1 (NOS1), increasing its expression in neuroblastoma cells. This effect can be further enhanced by all-trans-retinoic acid (ATRA) treatment, which promotes miRNA-mediated NOS1 gene demethylation and neural cell differentiation while inhibiting tumor growth [85].

In osteosarcoma cells, miR-29a-mediated DNMT3B repression decreases SOCS1 methylation, resulting in the upregulation of its expression [86]. SOCS1 is a tumor suppressor, and its silencing by hypermethylation of its promoter region is frequently seen in common pediatric malignancies such as leukemia and brain tumors [87,88,89]. Furthermore, proteins that bind to methylated DNAs to regulate chromatin organization and gene expression, including methyl-CpG binding protein 2 (MECP2) and methyl-CpG-binding domain 2 (MBD2), are direct targets of miRNAs [90,91]. MiR-454 binds to 3′-UTR of MECP2 and inhibits its expression in renal cell carcinoma (RCC) cells, the second most common type of kidney cancer in children. MECP2 inhibition is associated with suppression of RCC cell proliferation, migration, and invasion [92]. MBD2 is a target for miR-520b, and its miR-dependent inhibition facilitate tumor regression in gliomas [93].

MiRNA-mediated epigenetic gene silencing is not limited to DNA methylation, as studies have suggested that miRNAs can regulate N6-methyladenosine (m6A) marks on RNA molecules. M6A marks are the methylation at the sixth position of adenine of an RNA molecule (m6A), commonly found in both coding and noncoding RNAs and are widespread in cancers [94]. A large subset of mRNA transcripts has m6A sites in their untranslated (UTR) regions that harbor potential miRNA-binding regions. MiRNA binds to these regions and recruits methylation modifiers to regulate mRNA translation and protein expression. During glioma stem-cell (GSC) differentiation, AGO1 facilitates the binding of miR-145-5p to interleukin enhancer-binding factor 3 (ILF3), a double-stranded RNA-binding protein, and fat mass and obesity-associated protein (FTO), an RNA N6-methyladenosine (m6A) demethylase. This results in the recruitment of an FTO/AGO1/ILF3/miR-145 complex on CAP-Gly domain-containing linker protein 3 (CLIP3) mRNA. CLIP3 functions as a tumor suppressor in gliomas, and its downregulation promotes glioma radioresistance by enhancing stemness [95]. The FTO/AGO1/ILF3/miR-145 complex brings FTO close to CLIP3, inducing FTO-mediated m6A demethylation of CLIP3, which increases its translation during GSC differentiation [96].

Dysfunction of miRNAs has been associated with disturbed expression of oncogenic or tumor-suppressive target genes. Epigenetic regulation of miRNAs is essential for controlling normal miRNA functions and maintaining homeostasis. A growing number of miRNAs are found methylated in tumors, suggesting a critical role in DNA methylation to regulate miRNA-controlled gene expression at transcriptional levels. For instance, tumor suppressor miR-124-1 is frequently methylated and silenced in pediatric hematological malignancies, including AML and ALL [97]. Another tumor suppressor, mir-486-5p, is hypermethylated at CpG sites located in the promoter region of its host gene ANK1 in osteosarcoma patients [98]. Likewise, DNA hypermethylation in two CpG islands at the miR-449c locus inhibits miR-449c expression, increasing its target c-Myc and downstream signaling proteins and contributing to osteosarcoma tumorigenesis [99]. Furthermore, HOXB3 whose overexpression is associated with myeloproliferative neoplasm (MPN) disorders recruits DNMT3B to bind in the pre-miR-375 promoter. DNMT3B induces miR-375 DNA hypermethylation and silences its expression, promoting leukemogenesis [100].

Cancer-specific hypermethylation of miRNAs serves as an alternative mechanism to limit genes with tumor-suppressive functions and promote oncogenes at transcript levels. Thus, it is not surprising to find widespread hypermethylation of CpG islands in the regulatory regions of miRNA genes. Methylation-dependent miRNA repression is crucial in establishing gene expression patterns that help cancer cells acquire abnormal phenotypes during tumor development. Several miRNA genes associated with MLL or MLL fusions harbor CpG methyl marks in infant patients with MLL-rearranged acute lymphoblastic leukemia (ALL), the most aggressive type of childhood leukemia [101]. One example is the hypermethylation of tumor suppressor miR let-7b in refractory infant MLL, which decreases its expression and enhances leukemia cell growth, promoting leukemogenesis [102]. The differential methylation patterns at CpG sites in miRNA genes located on the 14q32 miRNA cluster, also known as the imprinted DLK1–MEG3 genomic region, are associated with the prognosis in patients with osteosarcoma. These methylation changes show a strong correlation with aggressive osteosarcoma cell behavior in vitro and can be indicative of osteosarcoma patients’ outcome.

In addition to CpG methylation, mature miRNAs can be methylated at 5-cytosine positions (m5C), which regulates their interactions with target mRNAs and their functions. The cytosine methylation of miR-181a-5p inhibits its antitumor functions in glioblastoma cells. Specifically, DNMT3A and AGO4 form a complex with mature miR-181a and induce its cytosine methylation. Cytosine-methylated miR-181a-5p loses its ability to interact with target 3′UTR mRNA, suppressing its antiproliferative and anti-invasion activity in glioblastoma cells [103]. Furthermore, methylation-dependent miRNA silencing may serve as a mechanism to supplement neoplastic progression due to gene deletion that causes loss of tumor suppressive functions. For instance, an increase in methylation of CpG islands located in the region upstream of miR-15a and miR-15b promoters supplements the loss of functions of these miRNAs due to their genetic deletion in AML cells [104]. This suggests that the loss of expression of miR-15a and miR-15b caused by genetic loss and methylation-mediated silencing may contribute to AML pathogenesis.

#### 2.2.2. LncRNAs

LncRNAs can interact with DNMTs and modulate their activity by recruiting DNMTs to target genes, blocking the access of DNMTs to the DNA, promoting DNMT degradation, or binding to polycomb group proteins such as EZH2 that can recruit DNMTs to target loci. The modulation of DNMTs by lncRNAs has been shown to play a role in tumorigenesis by affecting the methylation status and expression of target genes. For instance, lncRNA HOTAIR recruits Dnmt3b to the promoter region of tumor suppressor gene HOXA5, leading to its increased methylation and silencing in AML cells. HOTAIR downregulation upregulates HOXA5, which inhibits proliferation and induces apoptosis in AML cells [105]. LncRNA LAMP5-AS1 directly binds to methyltransferase DOT1L and promotes DOT1L-mediated H3K79 dimethylation and trimethylation of target genes involved in leukemia stem-cell proliferation. LAMP5-AS1 is highly expressed in patients with MLL leukemia than in other leukemia types. LAMP5-AS1-mediated DOT1L recruitment inhibits H3K79 methylation of the HOXA gene cluster, CDK6 and MEIS1, which are the core target genes of MLL fusion proteins. CDK6 and MEIS1 upregulation promotes MLL leukemia cell self-renewal and inhibits differentiation. This suggests that LAMP5-AS1 plays a critical role in developing and maintaining leukemia stem cells with MLL rearrangements [106].

Furthermore, lncRNA GALH promotes the proliferation and lung metastasis of HCC cells by controlling the methylation status of Gankyrin, a negative regulator of the CDK/Rb and HDM2/P53 tumor suppressor pathways. LncGALH induces DNMT1’s ubiquitination, decreasing Gankyrin gene methylation and increasing its expression [107]. Some lncRNAs act as scaffolds that recruit EZH2 and DNMTs to specific genomic regions, thereby promoting the repression of target genes. In HCC cells, lncRNA HOTAIR promotes DNMT1 binding to miR-122 gene promoter via EZH2, leading to miR-122 methylation and repression. Mir-122 downregulation contributes to the progression of HCC by promoting HCC cell proliferation and tumor growth. The interaction of HOTAIR with miR-122 via EZH2-mediated DNA methylation is a unique mechanism of lncRNA–miRNA interaction, contributing to gene regulation in tumor cells [108].

Moreover, studies have shown that promoter hypermethylation of lncRNAs is one of the most frequent pathways via which tumor cells evade standard regulatory mechanisms and promote tumor growth. For example, in diffuse large B-cell lymphoma (DLBCL), the lncRNA NKILA is often silenced through hypermethylation of its promoter region, which leads to increased cellular proliferation and decreased cell death. NKILA inhibits the phosphorylation of IκBα, which prevents its degradation and the subsequent nuclear translocation of both total and phosphorylated p65, a key component of the NF-κB complex. Repression of the NF-κB signaling pathway plays a critical role in regulating NKILA-mediated tumor suppressive functions [109].

Overall, these studies suggest that lncRNA-mediated epigenetic modifications, or vice versa, are critical events in cancer pathogenesis. Further understanding the molecular mechanisms associated with changes in epigenetic patterns of lncRNAs can provide important insights into the complex regulation of gene expression and its impact on cellular function and cancer development.

## 3. Role of ncRNAS in the Epigenetic Regulation of the TME

The TME is the complex and dynamic network of cellular components that surrounds cancer cells during tumor development and progression. The TME of a developing tumor is composed of cancer cells, infiltrating inflammatory and immune cells, stromal cells, blood vessels, and the extracellular matrix. There is constant communication between cancer cells and other TME components through various molecular signaling pathways mediated by ncRNAs, which can either promote or inhibit tumor growth. NcRNA-mediated communication within the cancer microenvironment plays a significant role in shaping the epigenetic landscape of cancer and TME cells, which can influence the tumor’s behavior. This section focuses on ncRNAs’ functions in the epigenetic regulation of TME cells and their role in tumor development and progression.

### 3.1. miRNAs

A tumor and the cells in its microenvironment regulate or exchange one another’s miRNAs to influence gene expression in the TME, which plays an essential role in tumor differentiation, development, and immune evasion. For instance, stromal cells residing in the microenvironment of mantle cell lymphoma (MCL) and other non-Hodgkin lymphoma cells downregulate miR-548m in cancer cells, thereupon upregulating its target gene HDAC6. Mechanistically, stromal cells induce *c-MYC* expression in lymphoma cells, which recruits EZH2 to miR-548m gene promoter and inhibits its expression. HDAC6 upregulation contributes to lymphoma cell survival, drug resistance, and tumor progression [110]. HDAC6 deregulation is a significant contributing factor to tumor development due to its involvement in regulating multiple cellular processes [111]. In addition to altering epigenetic modifications in cancer-associated genes, HDAC6 directly regulates various other cellular functions, including cell proliferation [112], cell cycle [113], apoptosis [114], invasion [115], and metastasis [116]. This implies that miR-548m regulation in cancer cells by the tumor stroma can disrupt epigenetic regulation in tumor cells and plays a causative role during tumor initiation and development.

MiRNAs secreted from tumor-associated macrophages (TAMs) display a similar effect of modulating tumor cell behavior and growth. M2-type TAMs deliver miR-155-3p to MB cells via exosomes to regulate WDR82, which recruits the Setd1A/histone H3-Lys4 methyltransferase complex. The latter facilitates histone H3K4me of target genes by recruiting chromatin remodelers SETD1A or SETD1B to the C-terminal domain (CTD) of RNA polymerase II large subunit (POLR2A). MiR-155-3p directly targets WDR82 in MB cells, which promotes cell invasion and migration, and supports tumor growth [117]. Moreover, miR-22 mimics transfection in macrophages, which decreases lipopolysaccharide (LPS)-induced activation of HDAC6 and its downstream NF-κB and AP-1, leading to an increase in proinflammatory cytokines such as TNF-α, IL-1β, and IL-6 [118].

Reciprocally, changes in miRNA expression in cancer cells trigger the recruitment and activation of TAMs. Once activated, TAMs can promote tumor growth and immune evasion by secreting proinflammatory cytokines and growth factors, suppressing the immune response. Induction of miR-302a in glioma cells by histone demethylase Jumonji domain-containing 1C (JMJD1C) promotes M1 macrophage polarization, which is associated with enhanced antitumor effects in gliomas [119]. Specifically, JMJD1C promotes H3K9 demethylation at miR-302a promoter regions and increases its expression in glioma cells. Upregulation of miR-302a decreases its target mRNA methylase, METTL3, inhibiting METTL3-mediated m6A methylation and degradation of SOCS2. The increase in SOCS2 expression promotes the M1 TAM population in glioma tumors, resulting in the suppression of glioma growth [119]. Likewise, ectopic expression of HDAC6 in HCC cells induces histone deacetylations on three promoter regions of miRNA let-7i-5p and inhibits its expression. Let-7i-5p downregulation in HCC cells upregulates its target TSP1, which interacts with CD47 cell surface receptors on endothelial cells and macrophages. The TSP1–CD47 interaction inhibits endothelial cell migration and proliferation, suppressing tumor angiogenesis. TSP1 disrupts the CD47–SIRPα interaction on macrophages, which ceases the “do not eat me” signaling between HCC and macrophage cells and promotes macrophage-mediated phagocytosis [120].

Furthermore, the miR-144/miR-451a cluster and EZH2 form a feedback circuit to regulate TAM polarization in HCC cells [121]. MiR-144 can target and inhibit EZH2, whereas EZH2 can repress miR-144/miR-451a expression through histone H3K27 methylation at their promoter regions. Overexpression of miR-144/miR-451a in HCC cells inhibits hepatocyte growth factor (HGF) and macrophage migration-inhibitory factor (MIF) secretion, which promotes M1-like and represses M2-like polarization in TAMs. This implies that the hypermethylation-mediated suppression of miR-144/miR-451a cluster in HCC cells acts as an epigenetic regulatory mechanism to modulate TAMs for supporting tumor growth [121].

Moreover, fibrotic transition during HCC progression inhibits the transcription of antifibrotic pri-miRNAs, including let-7-5p, miR-30-5p, miR-29c-3p, miR-335-3p, and miR-338-3p in hepatic stellate cells, either by suppressing their transcriptional regulator Pparγ or by promoting CpG hypermethylation at their promoter regions. Loss of these pri-miRNAs functions triggers rapid upregulation of their target profibrotic mRNAs, leading to enhanced fibrosis during HCC development [122]. Fibroblasts, which are prominent in the TME, facilitate tumor cell growth and promote inflammation, and are associated with fibrosis. These fibroblasts, also known as cancer-associated fibroblasts (CAFs), epigenetically regulate the miRNAs of tumor cells to modulate tumor growth. CAFs induce hypermethylation of miR-200b-3p promoter regions in gastric carcinoma cells, decreasing miR-200b, which promotes tumor cell invasion and migration [123].

In another mechanism, CAFs secrete miRNAs into extracellular vehicles (EVs) that are subsequently taken up by cancer cells and regulate their gene expression. For instance, overexpression of tumor suppressor miR-124 in CAFs promotes its uptake by neighboring cancer cells, where miR-124 directly targets chemokine (C-C motif) ligand 2 (CCL2) and interleukin 8 (IL-8). CCL2 and IL-8 downregulation inhibits proliferation, migration, and growth of oral carcinoma (OC) cells [124]. High promoter methylation of the miR-124 promoter in OC and CAFs acts as an inhibitory mechanism to suppress miR-124 activity during OC growth.

Another important component of the TME is the endothelial cells that line blood vessels and are critical in tumor angiogenesis [125]. MiRNA exchange between tumor and endothelial cells is essential in regulating angiogenesis in various tumors. For example, glioblastoma-derived EVs deliver miR-9-5p to human umbilical ECs (HUVEC), which induces aberrant blood vessel formation in tumors and promotes tumor growth [126]. Hypermethylation of mir-9 is seen in more advanced stages tumors, suggesting its epigenetic silencing as a potential tumor marker of poor outcome in various cancer types [127,128]. Furthermore, mRNAs play an important role in defining T-cell fate through their epigenetic programming in the TME. For instance, miR-155 prevents effector T-cell senescence and functional exhaustion by targeting signaling pathways involved in T-cell differentiation [129]. MiR-155 targets the Src homology 2 (SH2) domain-containing inositol polyphosphate 5-phosphatase 1 (SHIP1), a negative regulator of PI3K/Akt signaling, which upregulates Akt-dependent Phf19 transcription. Phf19 is a polycomb-like protein that recruits the PRC2 subunit EZH2 to the regulatory regions of genes involved in terminal differentiation. EZH2-mediated H3K27me3 of genes such as Id2, Eomes, Prdm1, Zeb2, Maf, and Nr4a2 inhibits their expression and prevents terminal differentiation of T cells [129]. This suggest that targeting EV miRNAs to reprogram the fate of cytotoxic CD8^+^ T cells to enhance their ability to fight tumors could be a promising strategy to develop more effective immunotherapy treatments.

Supporting these thoughts, delivery of miR-148a inhibitor (miR-148i) using nanoparticle-based therapeutics could effectively enhance the antigen-presenting efficiency of tumor-associated dendritic cells (TADCs) to T cells and develop effective immune responses to tumors. The treatment of the miR-148i vaccine made of miR-148i, poly I:C, and ovalbumin encapsulated into polypeptide micelles significantly enhances Toll-like receptor 3 (TLR3)-induced dendritic cell maturation and activation in mice bearing melanoma tumors. Mechanistically, miR-148a inhibition upregulates its target DNMT1, leading to hypomethylation and downregulation of SOCS1, the suppressor of TLR signaling [130]. The activation of TLR3 signaling induces maturation of DCs following antigen uptake and enhances DC’s ability to efficiently present internalized antigens on major histocompatibility complex I (MHC-I) molecules to T cells. In summary, miRNA-mediated epigenetic modifications are critical for bidirectional communication between the tumor and its microenvironment and can contribute to the malignant phenotype of cancer cells (Figure 1).

Furthermore, recent studies suggest that the T helper type 2 (Th2) or the type 2 immune response may play a role in modulating the generation of immunity to cancer [131]. Type 2 immune responses are characterized by the recruitment of eosinophils, mast cells, basophils, and type 2 innate lymphoid cells (ILC2) into inflammatory tissues, which is associated with the production of interleukin-4 (IL-4), IL-5, IL-13, and thymic stromal lymphopoietin (TSLP) [132]. The Th2-induced inflammatory response primarily functions to eliminate extracellular parasites, bacteria, allergens, and toxins. MiRNAs can regulate mast-cell development, eosinophil differentiation, and their functions to modulate allergic inflammation, which may contribute to cancer development and progression [133,134,135]. For instance, COX-2 targeting miR-26 prevents the induction of COX-2 and its downstream HDAC3 during allergic inflammation, which is associated with a decreased interaction between mast cells and macrophages. Mast cells and macrophages activated during allergic inflammation can interact with tumor cells and promote tumor angiogenesis. By regulating these interactions, miR-26a and miR-26b can potentially reduce the tumorigenic and metastatic potential of cancer cells that is enhanced by allergic inflammation [136].

Similarly, the downregulation of miR-218 and miR-181a upregulates their target transglutaminase 2 (TGaseII), which promotes the interaction between mast cells and macrophages during allergic inflammation. This interaction is associated with the enhanced metastatic potential of tumor cells that accompanies allergic inflammation. Overall, these findings suggest that the downregulation of miR-218 and miR-181a, and the resulting upregulation of TGaseII, may be a mechanism via which allergic inflammation promotes the metastasis of tumor cells [137]. Furthermore, many miRNAs are differentially expressed in ILC2 and regulate their functions during type 2 immunity. For example, miR-155 inhibits apoptosis in ILC2 cells and plays a critical role in promoting their survival during type 2 immune responses [138]. Other miRNAs involved in the regulation of ILC2 functions include miR-142 and the miR-17-92 cluster, which regulate various aspects of ILC2 activation, proliferation, and cytokine production [139,140].

### 3.2. LncRNAs

Like miRNAs, the exchange of lncRNAs between cancer cells and surrounding cells can impact the behavior of both, affecting tumor progression and response to treatment. LncRNA-induced epigenetic changes in the TME significantly impact tumor growth and the ability to metastasize to distant sites. For example, the overexpression of the lncRNA LNMAT1 in bladder cancer cells induces the recruitment of macrophages to the TME, promoting bladder cancer-associated lymphangiogenesis and lymphatic metastasis [141]. LNMAT1 directly interacts with hnRNPL, a type of RNA-binding protein, and recruits hnRNPL to the CCL2 promoter. HnRNPL occupancy induces H3K4 trimethylation at the CCL2 promoter, resulting in its transcriptional activation. CCL2 is a chemokine produced by cancer cells that attracts tumor-promoting macrophages to the tumor site [141]. Another lncRNA, PTPRE-AS1, regulates receptor-type tyrosine-protein phosphatase (PTPRE) activity to modulate IL-4-induced M2 macrophage activation [142]. IL-4 is an anti-inflammatory cytokine that promotes TAM polarization and has been associated with tumor progression and metastasis [143]. PTPRE-AS1 directly interacts with WD repeat-containing protein 5 (WDR5), a subunit of the SET1/MLL family of H3K4 methyltransferases, and recruits it to the PTPRE promoter. WDR5 recruitment induces PTPRE promoter H3K4 trimethylation, thereby transcriptionally activating its gene expression. PTPRE activation is associated with the inhibition of IL-4-induced M2 macrophage activation and the development of allergic pulmonary inflammation [142].

In addition, PTPRE overexpression exerts antioncogenic functions by negatively regulating tumor cell growth [144,145,146]. Furthermore, lncRNA Snhg6, which is highly expressed in tumor-derived myeloid-derived suppressor cells (MDSCs), promotes the differentiation of CD11b^+^ Ly6G^−^ Ly6C^high^ monocytic (M)-MDSCs by stabilizing EZH2 through the protein ubiquitination degradation pathway. The differentiation of immature MDSCs into macrophages and dendritic cells in the TME plays a major role in immunosuppression and immune evasion by the tumor. Abnormal expression of lncRNA Snhg6 is seen in pediatric tumors, including Wilms tumor [147], gliomas [148], osteosarcoma [149], and liver cancers [150], which is associated with sustained tumor growth and metastases. Thus, lncRNA Snhg6-mediated epigenetic deregulation in the TME cells could contribute to its oncogenic functions in various tumors.

LncRNAs have been shown to epigenetically modulate the differentiation and activation of CD8^+^ T cells, as well as their ability to migrate to the tumor site and generate an antitumor response. For instance, lncRNA metastasis-associated lung adenocarcinoma transcript 1 (MALAT1) regulates CD8^+^ T-cell differentiation into effector CTLs and long-lived, functional memory CD8^+^ T cells by specifically regulating memory cell-associated genes [151]. Mechanistically, MALAT1 interacts with EZH2 and catalyzes H3K27me3 mediated silencing of genes involved in memory cell differentiation [151]. MALAT1 overexpression in diffuse large B-cell lymphoma cells (DLBCL) limits effector CD8^+^ T-cell functions by suppressing their proliferation and inducing apoptosis [152]. This implies that MALAT1 exchange in the TME helps tumor cells establish a tumor-friendly immune response, providing a survival advantage in the stressful TME infiltrated with antitumor immune cells.

A recent comprehensive genome study characterizing the genotype–immunophenotype association between lincRNA expression and tumor immune response in 32 cancer types revealed distinct expression patterns of lncRNAs strongly correlated with antitumor immunity [153]. One such lncRNA EPIC1, which is overexpressed in multiple tumors, suppresses antigen presentation by the tumor cells by epigenetically regulating IFN-receptor IFNGR1 and antigen presentation genes (APGs). EPIC1 promotes EZH2 occupancy on IFNGR1 and APGs, inducing H3K27me3 on their promoter region, which leads to their repression. IFNGR1 downregulation inhibits its downstream IFN-γ–Janus kinase–STAT1 signaling, whereas APG downregulation decreases MHC-I expression [153]. Inhibition of IFN-γ via JAK2–STAT1 signaling enhances PD-L1 expression, which is associated with low T-cell activation and cytokine secretion of T cells [154]. This helps tumors to evade attacks from the immune system and develop resistance to immunotherapy [153].

This implies that lncRNAs can indirectly regulate immune checkpoint proteins in tumor cells by epigenetically modulating their upstream regulators. Any alterations in the lncRNA expression can affect the epigenetic landscape of the tumor or surrounding TME cells, leading to tumor development and progression by altering gene expression patterns and disrupting normal cellular processes (Figure 1). The rewiring of the cancer epigenome by lncRNAs provides a survival advantage for various cancers by contributing to different cancer hallmarks (Figure 2).

In addition, differential lncRNA signatures between tumor and adjacent normal cells, across different cancer subtypes, and between therapy responder and non-responder patients make them candidate biomarkers for the diagnosis and prognosis of different cancer types. Table 1 discusses some other lncRNA examples regulating the epigenetics of the TME and their potential as diagnostic/prognostic candidate biomarkers in pediatric tumors.

### 3.3. PiRNAs and snRNAs

PiRNAs and snRNAs are other noncoding transcript types that play an important role in regulating epigenetic modifications in cancer. PiRNAs are small, single-stranded, and 23–36 nucleotides in length; they form RNA–protein complexes with PIWI proteins [155]. PIWI proteins are mainly found in germ cells, and their interaction with piRNAs helps to regulate RNA silencing [156].

Most piRNAs are generated from piRNA clusters, which are specific genetic regions that give rise to single-stranded piRNA precursors [157,158]. These precursors are transported from the nucleus and processed into 5′-monophosphate-containing pre-piRNAs by RNA helicase Armitage and endonuclease Zucchini [159,160,161]. The pre-piRNAs are then loaded onto PIWI proteins, which facilitate piRNA trimming by the exonuclease Nibbler, followed by methylation at its 2′ oxygen by the small RNA 2′-O-methyltransferase Hen1 [162,163,164,165,166]. This methylation enhances piRNA stability and generates functional piRNAs that can be further amplified by Ago3 and Aubergine (Aub) proteins [167,168]. Once processed, piRNAs can regulate various cellular functions, including transposon silencing [169], mRNA silencing [170], and chromatin assembly [171].

SnRNAs, on the other hand, are RNA molecules that form complexes with various proteins, forming RNA–protein complexes known as small nuclear ribonucleoproteins (snRNPs) [172]. SnRNAs are about 100–300 nucleotides long and primarily localized in the nucleus. They are involved in RNA-processing events such as intron splicing [173] and RNA stability [173]. Studies have suggested that piRNAs and snRNAs can modulate DNA methylation or histone modifications, affecting chromatin structure and gene expression [174,175,176,177]. Aberrant expression of piRNAs and snRNAs in cancer can affect epigenetic regulation of gene expression and contribute to cancer development through various mechanisms. Table 2 outlines the role of these ncRNA types in epigenetic gene expression regulation in tumors, while Table 3 provides links to databases and tools for identifying noncoding and coding RNA interactions and expression analyses.

**Table 1 cancers-15-02833-t001:** LncRNAs and their role in modulating tumor microenvironment via epigenetic regulations.

LncRNAs	Cancer Type	Target Epigenetic Regulator	Mechanism of Action	Role as Potential Biomarker
HOX antisense intergenic RNA (HOTAIR) [178]	Cutaneous squamous cell carcinoma (CSCC)	Transcription factor Sp1	HOTAIR interacts and upregulates Sp1, promoting Sp1-induced DNMT1-mediated promoter methylation and repression of miR-199a. Downregulation of miR-199a promotes CSCC cell stemness and tumor progression.	Upregulated in CSCC tissues compared to normal adjacent cells and associated with worse patient prognosis.
LncRNA IRAIN [179,180]	RCC	Dnmt1, Dnmt3a, and Dnmt3b	IRAIN recruits Dnmt1, Dnmt3a, and Dnmt3b to the VEGFA promoter, inhibiting its expression. VEGFA downregulation inhibits ECs recruitment, tumor angiogenesis, and growth.	IRAIN has lower expression in RC tissues than in healthy renal tissues.
LINC00152 [181]	Gastric cancer	EZH2	LINC00152 recruits EZH2 to CXCL9 and CXCL10 promoters and epigenetically silences them. LINC00152 inhibition upregulates CXCL9, CXCL10, and CXCR3, which promotes intratumoral cytotoxic CD8^+^ T-cell infiltration.	LINC00152 is highly expressed in gastric cancer patients than normal counterparts.
Nuclear paraspeckle assembly transcript 1 (NEAT1) [182,183]	Glioblastoma	RNA-binding protein SFPQ	NEAT1 promotes paraspeckle assembly and relocates transcriptional repressor SFPQ from the CXCL8 promoter to paraspeckles, upregulating its protein product IL8 expression. IL8 secretion from tumor cells facilitates TAM recruitment and immunosuppression.	GBM tissues have higher expression of NEAT1 than low-grade glioma and normal brain tissues.
Colorectal neoplasia differentially expressed (CRNDE) [184,185]	HCC	p300/YY1 complex	CRNDE stabilizes the p300/YY1 complex and enhances histone H3K9 and H3K27 acetylation at the EGFR promoter, upregulating its expression. Exosomal EGFR is known to modulate the liver microenvironment to facilitate liver metastases.	CRNDE is highly expressed in human HCC compared to normal liver cells.
MIAT [186,187,188]	Thyroid cancer	EZH2	MIAT sponges miR-150 activity and upregulates its target EZH2, promoting tumor cell proliferation, migration, and invasion. MIAT upregulation is associated with immune suppression in cancer.	MIAT is overexpressed in thyroid cancer patients
HOTAIR [189]	AML	EZH2	HOTAIR recruits EZH2 to the p15 promoter, inducing its H3K27me3 and silencing gene expression. p15 downregulation is associated with the enhanced self-renewal capacity of leukemia stem cells, promoting leukemogenesis.	HOTAIR expression is significantly upregulated in AML patients.

**Table 2 cancers-15-02833-t002:** The role of piRNAs and snRNAs in the epigenetic regulation of tumors and their microenvironment.

NcRNA Types	Cancer Type	Target Epigenetic Regulator	Mechanism of Action
piRNA-823 [190]	Multiple myeloma	DNMT3A and 3B	piRNA-823 overexpression upregulates DNMT3A and DNMT3B levels and increases global DNA methylation. PiRNA-823 silencing reexpress methylation-silenced tumor suppressor, p16INK4A, decreases tumor angiogenesis, and inhibits tumor growth.
piR_011186 [191]	AML	DNMT1, Suv39H1 and/or EZH2	piR_011186 promotes DNA and histone H3 methylation of the CDKN2B promoter, which downregulates its expression and is associated with enhanced cell proliferation.
PIWI-like 4 (piRNA associated protein) [192]	Glioma	H3K27me3 demethylase UTX	PIWIL4 interacts with UTX, which removes transcriptionally repressive H3K27me3 marks on neuronal genes, promoting neuronal differentiation and activity. The upregulation of neuronal genes due to PIWIL4–UTX interaction can further modify the glioma microenvironment and promote glioma cell proliferation.
piRNA-823 [193]	Multiple myeloma	DNMT3B	G-MDSCs induce piRNA-823 expression in multiple myeloma cells, which in turn activates DNMT3B expression and increases global DNA methylation. These changes are associated with enhanced stemness of multiple myeloma stem cells and tumor growth.
piRNA-30473 [194]	DLBCL	m6A mRNA methylase WTAP	piRNA-30473 stabilizes WTAP mRNA, upregulating global m6A levels in DLBCL cells. This increases hexokinase 2 (HK2) expression, which is associated with increased cell proliferation and tumorigenicity of DLBCL cells.
sdnRNA-3 [176]	Melanoma	Chromatin-remodeling regulator Mi-2β	sdnRNA-3 promotes the enrichment of chromodomain-helicase-DNA-binding protein 4 (CHD4), also known as Mi-2β, to the Nos2 promoter. This induces H3K27me3 modification of the Nos2 gene and represses the transcription of its gene product, inducible nitric oxide synthase (iNOS). The decrease in sdnRNA-3 expression in TAMs increases iNOS transcription and inhibits tumor growth.
RN7SK [177]	Multiple cancers including lung, liver, colon, and gastric	m6A readers	M6A readers recognize and interact with m6A-modified RN7SK, which facilitates the formation of RN7SK secondary structures and stabilizes its expression. RN7SK prevents the mRNA degradation of m6A readers by exonucleases, increasing their expression. The upregulation of m6A readers such as EWSR1 and KHDRBS1 promotes Wnt/β-catenin signaling and tumorigenesis by suppressing ubiquitin protein Cullin1 in various tumor types.
U1 snRNP [195]	Lung and breast cancer	Proximal polyadenylation signals (PASs) in introns and exons	U1 snRNP inhibits proximal polyadenylation signals (PASs) in introns and last exons, preventing premature transcription termination and mRNA shortening of target genes. U1 snRNP inhibition in cancer cells prompts the removal of 3ʹUTR miRNA target sites from many oncogenic mRNAs, resulting in their upregulation. The upregulated genes are involved in signaling pathways that control cell-cycle progression (CDC25A, CCNB1, and BRCA1), apoptosis (BCL6), cell growth (FGFR1, EGFR, and BRAF), cell migration (FGFR1, FYN, and TIMP2), extracellular matrix remodeling (TIMP2), DNA replication (APC), and transcription (EWSR1).

CDKN2B: cyclin-dependent kinase inhibitor 2B, G-MDSCs: granulocytic MDSCs, DLBCL: diffuse large B-cell lymphoma, sdnRNA-3: small nuclear (snRNA)/ small nucleolar RNA (snoRNA)-derived nuclear RNAs (sdnRNAs), m6A: N6-methyladenosine, EWSR1: EWS RNA-binding protein 1, KHDRBS1: KH RNA-binding domain-containing, signal transduction-associated 1, BRCA1: breast cancer gene 1, TIMP2: tissue inhibitor of metalloproteinases 2, APC: adenomatous polyposis coli.

**Table 3 cancers-15-02833-t003:** List of databases and common tools that can be used for identifying noncoding and coding RNA interactions and expression analyses.

Database Name	Used for	Link
miRBase [196]	This database contains information about miRNAs including sequences, annotations, and expression data. It also provides tools for predicting miRNA targets.	https://www.mirbase.org (accessed on 26 April 2023)
TargetScan [197]	This provides computational predictions of miRNA targets based on the matching of miRNA seed sequence with complementary mRNA sequences.	www.targetscan.org (accessed on 26 April 2023)
miRTarBase [198]	This database provides information about experimentally validated miRNA–target mRNA interactions.	www.mirtarbase.cuhk.edu.cn/ (accessed on 26 April 2023)
miRWalk [199]	miRWalk provides predictions of miRNA–target interactions based on several existing miRNA–target prediction programs, including TargetScan, miRanda, miRBase, and miRDB4. It also integrates these predictions with experimentally validated interactions from other databases, such as miRTarBase.	https://mirtarbase.cuhk.edu.cn/ (accessed on 26 April 2023)
NONCODE [200]	This comprehensive database contains information about expression and functions of lncRNA.	www.noncode.org/introduce.php (accessed on 26 April 2023)
lncRNAdb [201]	This database includes lncRNA annotations, functions, and interactions with other molecules.	https://ngdc.cncb.ac.cn/ (accessed on 26 April 2023)
LNCipedia [202]	It includes comprehensive information on lncRNA structure, sequence, expression, and functional annotation.	https://lncipedia.org/ (accessed on 26 April 2023)
LncBook [203]	It is a comprehensive dataset for studying the functions and mechanisms of lncRNAs. It integrates multi-omics data from expression, methylation, genome variation, and lncRNA–miRNA interactions, providing a more complete picture of the lncRNA molecular networks.	https://ngdc.cncb.ac.cn/lncbook/ (accessed on 26 April 2023)
LncATLAS [204] and LncLocator [205]	LncATLAS is a database of lncRNA subcellular localization, whereas LncLocator can predict the subcellular localization of the lncRNAs.	https://lncatlas.crg.eu/ (accessed on 26 April 2023) http://www.csbio.sjtu.edu.cn/bioinf/lncLocator/ (accessed on 26 April 2023)
piRNABank [206]	It provides information on piRNA sequences, genomic locations, expression patterns, and potential targets.	http://pirnabank.ibab.ac.in/ (accessed on 26 April 2023)
piRBase [207]	The database comprises piRNA sequences, genomic locations, expression patterns, targets, and functions.	http://bigdata.ibp.ac.cn/piRBase/ (accessed on 26 April 2023)
Rfam [208]	This database includes a variety of RNA families, including small nuclear RNAs. It provides annotation and alignment data, secondary structure predictions, and functional information for each family.	https://rfam.org/ (accessed on 26 April 2023)
snOPY [209]	It provides comprehensive information about small nucleolar RNA (snoRNAs), snoRNA gene loci, and target RNAs.	http://snoopy.med.miyazaki-u.ac.jp/snorna_db.cgi (accessed on 26 April 2023)
RNAcentral [210]	It is a comprehensive database that provides a single access point to a large and diverse collection of RNA sequences and their functions.	https://rnacentral.org/ (accessed on 26 April 2023)
starBase [211]	This database can be used to identify the RNA–RNA and protein–RNA interaction networks.	http://starbase.sysu.edu.cn/ (accessed on 26 April 2023)
RSEM [212]	This tool can be used for quantifying gene and transcript expression levels from RNA-Seq data.	software package
miRDeep2 [213]	This tool can be used to identify novel and known miRNAs in deep sequencing data.	software package
ANNOVAR [214]	This tool can be used to functionally annotate genetic variants, including noncoding regions detected from diverse genomes.	https://annovar.openbioinformatics.org/ (accessed on 26 April 2023)

**Figure 2 cancers-15-02833-f002:**
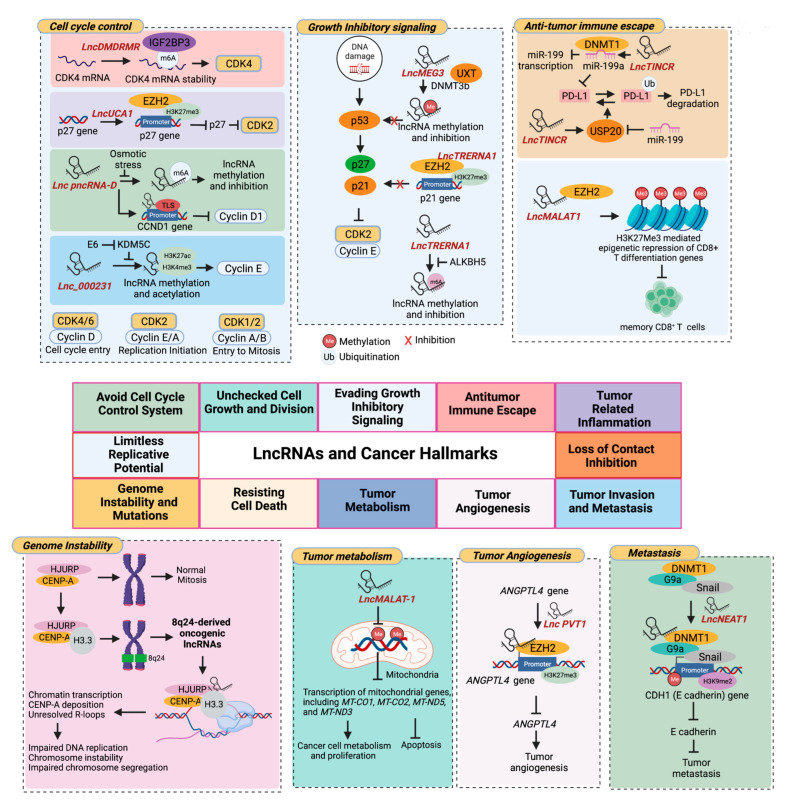
LncRNA-mediated epigenetic regulation of cancer hallmarks. **1. Evading Cell cycle control and unchecked cell division**: The different cyclin-CDK complexes regulate the orderly progression of the cell cycle by controlling key events such as DNA replication, mitosis, and cell division. If the rate of DNA damage exceeds cell’s capacity to fix it, these checkpoints inhibit cyclin-CDK complexes, preventing the accumulation and propagation of genetic errors during cell division. Many CDKS are regulated by lncRNAs, which modulate their expression during the cell cycle and DNA damage. For instance, oncogenic lncRNA DMDRMR promotes the binding of m6A reader insulin-like growth factor 2 mRNA-binding protein 3 (IGF2BP3) to the m6A-modified site located in the 5′ UTR of the CDK4 mRNA. This enhances the stability and translation efficiency of the CDK4 mRNA, leading to cell cycle progression and proliferation in RCC cells [215]. Another lncRNA, UCA1, associates with EZH2 and promotes the EZH2-mediated H3K27me3 of the p27 promoter, suppressing its expression. UCA1 overexpression in HCC cells inhibits p27, which upregulates CDK2 and promotes cell growth by facilitating the G1/S transition [216]. Furthermore, in normal cells, lncRNA pncRNA-D, which is transcribed from the promoter region of the CCND1 (cyclin D1) gene, is heavily m6A-methylated. Methylation affects pncRNA-D stability and promotes its degradation. In addition, m6A modification of pncRNA-D is recognized by nuclear YTHDC1, which prohibits its interaction with an RNA-binding protein TLS. Osmotic stress or irradiation decreases pncRNA-D methylation and increases its stability. This allows it to recruit TLS to the promoter of CCND1 gene, which leads to the reduction of cyclin D1 expression and induces cell cycle arrest [217]. Moreover, lnc_000231 is regulated by the E6 protein in cervical cancer cells. E6 interacts with histone demethylase KDM5C and promotes its degradation. This increases H3K27ac and H3K4me3 modification of the lnc_000231 promoter and upregulates its expression. Lnc_000231 sponges miR-497-5p activity, upregulating its target cyclin E1 (CCNE1) expression and promoting cervical cancer progression [218]. These findings suggest that lncRNA epigenetics are crucial in regulating the cell cycle, and lncRNAs deregulation can either promote or inhibit cell cycle progression in cancer. **2. Evading growth inhibitory signaling:** The tumor suppressor protein p53 plays a critical role in preventing cancer development. P53 governs the decisions of cells to proliferate or activate senescence and apoptotic programs in case of DNA damage. P53 activation induces a cyclin-dependent kinase inhibitor, p21, that can inhibit the activity of the CDK2-cyclin E complex. Many lncRNAs modulate p53 activity in cancer cells. For instance, MEG3 binds to p53 and positively regulates its transcription activity However, in breast cancer cells, ubiquitously expressed transcript (UXT) overexpression recruits DNMT3b to the MEG3 imprinting control region, leading to its hypermethylation and decreased expression. This decreases P53 and promotes tumor growth [219]. Another lncRNA, TRERNA1, regulates p21 expression by inducing EZH2-mediated H3K27me3 modification in its promoter region, thus epigenetically silencing its expression. In DLBCL cells, upregulation of m6A demethyltransferase ALKBH5 prevents m6A modification of TRERNA1 transcripts, leading to increased expression. TRERNA1 inhibits p21 and promotes proliferation and growth of DLBCL tumors [220]. **3. Antitumor immune escape:** Programmed death-ligand 1 (PD-L1) suppress antitumor immunity by inducing tumor-specific CD8+ T cell exhaustion [221]. PD-L1 is overexpressed expressed on the surface of some cancer cells. PD-L1 on cancer cells interacts with its receptor PD-1 on the surface of T cells, which can inhibit their activity and prevent anti-tumor immunity [222]. PD-L1 is a target of many lncRNAs in various cancer types. For example, the lncRNA TINCR upregulates PD-L1 expression in breast cancer cells, which impairs the effectiveness of anticancer immunotherapy. Mechanistically, TINCR recruits DNMT1 to PD-L1 and deubiquitinase USP20, targeting miRNA miR-199a-5p. This leads to miR-199a-5p methylation and suppression, upregulating USP20 and PD-L1 expression. USP20 inhibits PD-L1 ubiquitination, which increases its expression and prevents anti-tumor immunity [223]. Another lncRNA, MALAT-1, interacts with EZH2 to maintain its deposition on genes associated with memory cells in terminal effector (TE) CD8+ T cells. This results in increased deposition of H3K27me3 on these genes, which represses their activity and prevents memory cell differentiation [151]. **4. Tumor metastasis:** The loss of E-cadherin promotes the disaggregation of cancer cells from the tumor mass, which is the first step of the metastatic cascade [224]. LncRNA NEAT regulates tumor metastasis in OS cells by epigenetically regulating E-cadherin. NEAT1 interacts with the G9a-DNMT1-Snail complex and recruits it to the CDH1 (E-cadherin) promoter, which increases H3K9me2 and DNA methylation and inhibits E-cadherin expression. Overexpression of NEAT1 in osteosarcoma cells induces epithelial-mesenchymal transition (EMT) and promotes metastasis [225]. **5. Tumor angiogenesis:** LncRNAs regulate proteins involved in tumor angiogenesis. For instance, LncRNA PVT1 binds to EZH2 and epigenetically regulates angiopoietin-like 4 (ANGPTL4), which has been shown to regulate angiogenesis in various tumor types [226]. PVT1-mediated EZH2 binding to ANGPTL4 promoter increases H3K27me3 levels, which suppresses its expression [227]. **6. Tumor metabolism:** Mitochondria is the energy-generating organelle of the cell. LncRNAs have been shown to epigenetically regulate mitochondrial genes and modulate mitochondria metabolism in tumor cells. In HCC, lncRNA MALAT1 has been found to localize to the mitochondria, where it binds to mitochondrial DNA (mtDNA) and alters mtDNA methylation patterns. MALAT1 knockdown induces alterations in the mitochondrial transcriptome and decreases the expression of mitochondrial genes MT-CO1, MT-CO2, MT-ND5, and MT-ND3. This is associated with altered mitochondrial structure, low OXPHOS, decreased ATP production, reduced mitophagy, increased ROS production, and mitochondrial apoptosis in HCC cells [228]. **7. Genome instability and mutations:** Many locus-specific oncogenic lncRNAs have been found to alter the local chromatin landscapes, which can potentially induce chromosomal breaks during segregation, compromising chromosome integrity and leading to genomic instability. During the cell cycle, histone H3 variant CENP-A accumulates at the centromeres to ensure proper chromosome segregation. The deposition of CENP-A at centromeres involves a chaperone protein HJURP. CENP-A overexpression in cancer cells forms hybrid nucleosomes with H3.3 and hijacks the H3.3 chaperone pathway. CENP-A-H3.3 hybrid nucleosomes accumulate at chromosome sites such as the 8q24 locus, which alters the local chromatin landscape. The 8q24-derived oncogenic lncRNAs serve as a recruitment signal for incorrect chaperone-histone variant complexes, promoting further ectopic deposition of CENP-A. This leads to active transcription of the local chromatin, resulting in a higher R-loop level. The unresolved R-loop configuration with the ectopic CENP-A nucleosomes can impair replication efficiency, resulting in under-replicated DNA with stalled replication forks [229].

## 4. Conclusions

In the context of pediatric tumors, the role of epigenetic modifications in the development and progression of the disease is significant due to the low tumor mutational burden. A growing number of studies have shown the role of hundreds of ncRNAs in regulating epigenetic modifications affecting target mRNA expression in various tumor types. Conversely, epigenetic modifications can also affect ncRNA expression, and an intricate balance between ncRNAs and the epigenetic machinery helps maintain gene expression and cellular function (Figure 1). This balance is often found disturbed in pediatric tumors, resulting in increased epigenetic alterations affecting gene regulation and promoting disease pathogenesis. Numerous studies have highlighted the potential prognostic and therapeutic relevance of altered epigenetic profiles of ncRNAs in various pediatric tumor types. However, despite tremendous progress, especially in the past decade, the potential implications of ncRNA-based treatments for pediatric tumors are far from reality. One of the primary reasons for this is the genetic differences between primary and adult tumors, as pediatric tumors display different genetic mutations and epigenetic profiles than adult tumors, which can affect their response to treatment.

Other reasons include the limited experimental data available for most ncRNAs in pediatric tumors compared to adult tumors, the low specificity of ncRNAs in particular cancer types, as the same ncRNA behaves differently in two different cancer types, the lack of effective clinically approved delivery vehicles for ncRNA-based cancer therapies, limited clinical trials due to the smaller number of pediatric cases, and high amounts of heterogeneity in pediatric tumors. Since most pediatric tumors are epigenetically deregulated, a comprehensive understanding of ncRNAs as upstream regulators of key epigenetic signaling may provide new targets for therapeutic intervention. Although we are still far from clinically approved therapeutics targeting ncRNA-based epigenetic modifications, ongoing research has great potential in pediatric cancer therapeutics. This is opening up new opportunities in the future of cancer therapy.

## Figures and Tables

**Figure 1 cancers-15-02833-f001:**
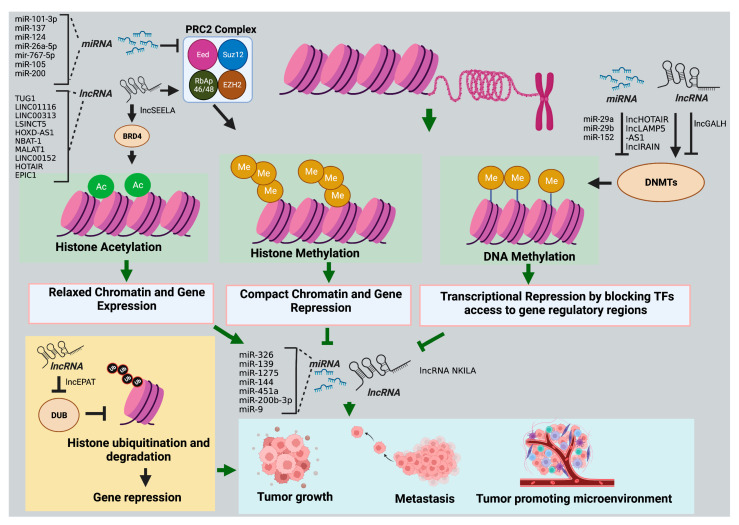
The diagram depicts the regulation of histone and DNA modifications by ncRNAs, including histone acetylation, histone methylation, and DNA methylation, which affect changes in DNA accessibility. Conversely, these epigenetic modifications regulate the ncRNAs, affecting their expression profiles in different tumor cells and playing a critical role in cancer development and progression. MiRNAs and lncRNAs can regulate the epigenetic multiprotein complex PRC2 and the conserved family of cytosine methylases DNMTs, which control epigenetic modifications. LncRNAs can recruit histone acetyltransferases (HATs) like BRD4 to promote histone acetylation, leading to target gene activation. LncRNAs can also inhibit deubiquitinating enzymes (DUBs), promoting histone degradation and gene repression. These epigenetic modifications can regulate the expression of miRNAs and lncRNAs, contributing to the development of cancer. Epigenetic aberrations, such as those affecting the regulation of ncRNAs, can promote genetic instability, leading to tumor development, a tumor-favorable microenvironment, and metastasis. The article discusses various mechanisms by which these epigenetic modifications and ncRNA dysregulation can promote growth in pediatric tumors. Overall, the diagram and accompanying text describe the complex interplay between epigenetic modifications, ncRNAs, and cancer development, highlighting the importance of understanding the role of epigenetics in tumor cell biology.

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
