# Peer review of "Crosstalk between Noncoding RNAs and the Epigenetics Machinery in Pediatric Tumors and Their Microenvironment"

_cancers, 2023, doi:10.3390/cancers15102833_

Round 1

Reviewer 1 Report

·      The author penned an interesting review of the interaction between non-coding RNAs and the epigenetic machinery in pediatric tumors and their microenvironment.

·      The literature used for the review article is good and updated.

·      The author must be careful of the spelling and other grammatical errors in the manuscript.

·      If possible, the author may use the full name of miRs (e.g., miR-145-5p).

·      Figure 1 quality is good, the author may use the name of specific miR and lncRNA instead of only miR and lncRNA

·      The discussion part should be more specific.

Author Response

Comments and Suggestions for Authors

  • The author penned an interesting review of the interaction between non-coding RNAs and the epigenetic machinery in paediatric tumours and their microenvironment.
  • The literature used for the review article is good and updated.

Reply: I am highly grateful to the reviewer for the thorough and critical review of this manuscript and for giving comments. The changes are highlighted in red.

  • The author must be careful of the spelling and other grammatical errors in the manuscript.

Reply: I want to thank the reviewer for pointing out the spelling and other grammatical errors in the manuscript. In the revised manuscript, the following errors have been corrected.

  • If possible, the author may use the full name of miRs (e.g., miR-145-5p).

Reply: Some papers cited in this review did not disclose the full name of the miRs they used for the study. Those who reported are included in the revised manuscript.

  • Figure 1 quality is good; the author may use the name of specific miR and lncRNA instead of only miR and lncRNA

Reply: As per reviewer suggestions, the name of specific miR and lncRNA are included in the revised manuscript.

  • The discussion part should be more specific.

Reply: In the revised manuscript, I provided more specific discussions.

Reviewer 2 Report

In the present manuscript, based on the available literature, the authors reviewed Non-Coding RNAs and the Epigenetics Machinery in Pediatric Tumors and their Microenvironment.

  Comments

 1. The introduction could be more focused and cover the topic comprehensively.

 2. The information is mostly a summarization of the available literature. However, it only gives very general information about the function and mechanism of action of the ncRNAs and their subsets. The functional distinction and uniqueness among various subsets are not prominent in the manuscript.

3. It could be interesting if the role of ncRNAs in the different stages of cancer could be incorporated.

4. What could be the effect of ncRNAs on Type- 2 (TH2) inflammatory responses?

5. The manuscript is often difficult to read. There are many typos in the manuscript. In addition, very long sentences are used that make it difficult to understand the message. In many places, the text contains a list of descriptions of separate studies that could be presented and summarized more comprehensively.

6. A flowchart or tables that represent available databases and common tools that use for identifying noncoding and coding RNA interactions and expression analyses should be included in detail to improve the reader's comprehension.

 7. More relevant and updated references should be added to the reference section.

Author Response

Comments and Suggestions for Authors

In the present manuscript, based on the available literature, the authors reviewed Non-Coding RNAs and the Epigenetics Machinery in Pediatric Tumors and their Microenvironment.

  Comments

  1. The introduction could be more focused and cover the topic comprehensively.

Reply: I want to thank the reviewer for their thoughtful comments and suggestions for improving this manuscript. I have revised the introduction section of the manuscript to provide a more comprehensive overview of ncRNAs. The changes are highlighted in red in the revised manuscript.

  1. The information is mostly a summarization of the available literature. However, it only gives very general information about the function and mechanism of action of the ncRNAs and their subsets. The functional distinction and uniqueness among various subsets are not prominent in the manuscript.

Reply: The review article aims to provide a comprehensive overview of the current understanding of ncRNA and its role in the epigenetic regulation of pediatric tumors. However, there is indeed a significant gap in research studies of pediatric tumors compared to adult tumors. One of the main reasons for this is that childhood cancers are relatively rare, which can make it more difficult to acquire research funding and recruit large patient cohorts for clinical studies. Therefore, there are not many studies in pediatric tumors that comprehensively describe the role of ncRNA-based epigenetic mechanisms in tumor development.

However, the review article attempts to summarize most studies that have elucidated ncRNA mechanisms and their functions in pediatric tumors to better understand their role in cancer and open new therapeutic possibilities to modulate their function. By providing an overview of the current understanding of ncRNA and its role in the epigenetic regulation of pediatric tumors, the review article may help to stimulate further research in this important area and promote the development of more effective and targeted therapies for children with cancer.

  1. It could be interesting if the role of ncRNAs in the different stages of cancer could be incorporated.

Reply: As per reviewer suggestions, I included figure.2 and its legends highlighting the role of lncRNA-mediated epigenetic regulation of cancer hallmarks.

  1. What could be the effect of ncRNAs on Type- 2 (TH2) inflammatory responses?

Reply: NcRNAs can regulate Type-2 (TH2) inflammatory responses. For instance, miRNAs can regulate mast cell development, eosinophil differentiation, and their functions to modulate Type- 2 (TH2) inflammatory responses, which may contribute to cancer development. These examples are discussed in the revised manuscript under the section: Role of ncRNAs in the epigenetic regulation of the TME (Last paragraph of the subsection: miRNAs)

  1. The manuscript is often difficult to read. There are many typos in the manuscript. In addition, very long sentences are used that make it difficult to understand the message. In many places, the text contains a list of descriptions of separate studies that could be presented and summarized more comprehensively.

Reply: I am grateful to the Reviewer for pointing out the spelling and other grammatical errors in the manuscript. The errors are corrected, and sentences are revised to improve clarity and readability. In addition, the studies are organised in a logical flow to help readers better understand the main concepts. I believe that these changes have strengthened the manuscript and improved its overall readability.

  1. A flowchart or tables that represent available databases and common tools that use for identifying noncoding and coding RNA interactions and expression analyses should be included in detail to improve the reader's comprehension.

Reply: As suggested by the reviewer, list of databases and common tools that can be used for identifying noncoding and coding RNA interactions and expression analyses are included in the revised manuscript (Table.3).

  1. More relevant and updated references should be added to the reference section.

Reply: The relevant and updated references are included in the reference section.

Reviewer 3 Report

This review focuses on the role of ncRNA in regulating the epigenetics of pediatric tumors and their tumor microenvironment. This is an important topic, but I found lack of literature support/survey in the current manuscript.

1. I thought the paper focused on Pediatric Tumors. However, many of the references and examples listed for both “epigenetic regulation of pediatric tumors” and “epigenetic regulation of the TME” are not from children study. I found many references working on cell lines and adult samples. Somehow it's quite confusing. I am not sure whether the described findings are derived from children study or by guessing. 

2. Other types of nRNAs (piRNAs, snRNA and endogenous siRNAs) are not discussed. 

3. A Figure about the relationship between cancer hallmarks and the role of ncRNA in epigenetic regulation would better help understand the concept.

4. Any data support the role of ncRNA in histone sumoylation and phosphorylation?

Author Response

Comments and Suggestions for Authors

This review focuses on the role of ncRNA in regulating the epigenetics of pediatric tumors and their tumor microenvironment. This is an important topic, but I found lack of literature support/survey in the current manuscript.

  1. I thought the paper focused on Pediatric Tumors. However, many of the references and examples listed for both “epigenetic regulation of pediatric tumors” and “epigenetic regulation of the TME” are not from children study. I found many references working on cell lines and adult samples. Somehow, it's quite confusing. I am not sure whether the described findings are derived from children study or by guessing. 

Reply: I am grateful to the reviewer for their thoughtful comments and suggestions on our manuscript. Most studies this paper discusses focus on pediatric tumor cell lines or samples from pediatric patients. To address this, I have added comments on each study included in the manuscript, clarifying the cell lines or patient samples used in the study. However, for some tumor types, such as glioblastoma, the studies discussed are on adult tissues because most glioma research has been conducted using adult glioblastoma cell lines. This is partly due to the rarity of pediatric glioblastoma, which accounts for only 10-15% of all childhood brain tumors (PMID: 35565425).

Although adult tumors may not ideally mimic the pediatric tumor microenvironment, they can provide valuable insights into cancer biology and help inform the development of new therapies. This is particularly important for very rare pediatric tumors such as childhood bladder, oral, and gastric cancer, where research is lacking. Therefore, studying adult tumors as models to understand some rare children's tumors can be a useful approach in some cases.

  1. Other types of ncRNAS (piRNAs, snRNA and endogenous siRNAs) are not discussed. 

Reply: As suggested by the reviewer, piRNAs and snRNAs and their role in the epigenetic regulation of tumors and their microenvironment are discussed in the revised manuscript (Table.2).

  1. A Figure about the relationship between cancer hallmarks and the role of ncRNA in epigenetic regulation would better help understand the concept.

Reply: As suggested by the reviewer, lncRNA-mediated epigenetic regulation of cancer hallmarks is discussed in the revised manuscript (figure.2 and its legends).

   4. Any data support the role of ncRNA in histone SUMOylating and phosphorylation?

Reply: Some miRNAs, such as MicroRNA-133b, can target SUMO1, but their role in the regulation of histone SUMOylation is not well-defined. However, several recent studies have reported the role of lncRNAs in histone phosphorylation in cancer. These studies are discussed in the revised manuscript under the section: Role of ncRNAs in the Epigenetic Regulation of Pediatric Tumors; subsection: Histone Modification and subsection: LncRNAs.